# A Quality Assurance Discrimination Tool for the Evaluation of Satellite Laboratory Practice Excellence in the Context of the European Official Meat Inspection for *Trichinella* spp.

**DOI:** 10.3390/foods12224186

**Published:** 2023-11-20

**Authors:** José Villegas-Pérez, Francisco Javier Navas-González, Salud Serrano, Fernando García-Viejo, Leandro Buffoni

**Affiliations:** 1Ph.D. School, University of Cordoba, 14011 Cordoba, Spain; josevillegasperez@gmail.com; 2Department of Animal Health, Area of Parasitology, Faculty of Veterinary Sciences, University of Cordoba, 14071 Cordoba, Spain; h12bupel@uco.es; 3Department of Genetics, Faculty of Veterinary Sciences, University of Cordoba, 14071 Cordoba, Spain; 4Department of Food Science and Technology, Faculty of Veterinary Sciences, University of Cordoba, 14071 Cordoba, Spain; bt2sejis@uco.es; 5Health and Family Counseling, Junta de Andalucia, 14004 Cordoba, Spain; fernando.garcia.viejo@juntadeandalucia.es

**Keywords:** Trichinellosis, *Trichinella*, laboratories, food safety, quality management system, canonical discriminant analysis

## Abstract

Trichinellosis is a parasitic foodborne zoonotic disease transmitted by ingestion of raw or undercooked meat containing the first larval stage (L1) of the nematode. To ensure the quality and safety of food intended for human consumption, meat inspection for detection of *Trichinella* spp. larvae is a mandatory procedure according to EU regulations. The implementation of quality assurance practices in laboratories that are responsible for *Trichinella* spp. detection is essential given that the detection of this parasite is still a pivotal threat to public health, and it is included in list A of Annex I, Directive 2003/99/EC, which determines the agents to be monitored on a mandatory basis. A Quality Management System (QMS) was applied to slaughterhouses and game handling establishments conducting *Trichinella* spp. testing without official accreditation but under the supervision of the relevant authority. This study aims to retrospectively analyze the outcomes of implementing the QMS in slaughterhouses and game handling establishments involved in Trichinella testing in southern Spain. Canonical discriminant analyses (CDAs) were performed to design a tool enabling the classification of SLs while determining whether linear combinations of measures of quality-assurance-related traits describe within- and between-SL clustering patterns. The participation of two or more auditors improves the homogeneity of the results deriving from audits. However, when training expertise ensures that such levels of inter-/intralaboratory homogeneity are reached, auditors can perform single audits and act as potential trainers for other auditors. Additionally, technical procedure issues were the primary risk factors identified during audits, which suggests that they should be considered a critical control point within the QMS.

## 1. Introduction

Trichinellosis is a worldwide foodborne zoonotic disease caused by the ingestion of the helminth *Trichinella* spp. Pigs, both domestic and wild, are the main reservoirs, and human infection is primarily linked to the consumption of raw or undercooked meat from infected animals without veterinary inspection [1]. Currently, this parasite continues to pose a significant threat to public health, and it is included in list A of Annex I, Directive 2003/99/EC [2] on the surveillance of zoonoses and zoonotic agents, which determines the agents that have to be monitored on a mandatory basis. According to the European Union (EU), 117 human cases have been reported during 2020, with 99 out of the 117 cases acquired within the EU [3].

The detection of *Trichinella* spp. larvae during meat inspection is performed using the magnetic stirrer artificial digestion technique. The technique is considered the “gold” standard method, being “capable of consistently detecting *Trichinella* larvae in meat at a level of sensitivity that is recognized to be effective for use in controlling animal infection and preventing human disease” [4]. Also, it is a mandatory procedure according to EU regulation (EU 2015/1375 [5]), which lays down specific rules on official controls for *Trichinella* in meat. This regulation also specifies equivalent techniques for *Trichinella* spp. larvae detection in analyzed samples.

As the International Commission on Trichinellosis exposes [4], the implementation of quality assurance practices in laboratories performing *Trichinella* spp. detection is of the utmost importance, and the probability of detecting a positive case, if present, must be high (>95 or >99%) in order to ensure that there is a low or negligible risk of transmission to humans through the food chain [6]. Laboratories accredited according to ISO/IEC 17025 [6] for *Trichinella* digestion testing are required to use validated diagnostic methods to confirm that the methods are fit for the intended use. In 2019, the International Commission on Trichinellosis recommended the adoption of system-wide practices for quality assurance [4]. However, the minimum required standards for quality assurance are determined and implemented by each local public health authority in EU member states. European regulation (EU 625/2017) [7] on official controls and other official activities performed to ensure the application of food and feed law and rules on animal health and welfare, plant health, and plant protection products states that the official control of *Trichinella* spp. larvae presence during meat inspection should be conducted in accredited laboratories designated by the competent authority. Furthermore, laboratories solely engaged in Trichinella spp. detection in meat using the methods outlined in EU 2015/1375 [5] may be exempt from accreditation if they operate under the supervision of the competent authority.

The activities performed in internal laboratories require continuous analysis and evaluation by the competent authority. Assessing critical control points involves various aspects such as personal training, diagnostic procedure performance, and document registration, among others. Additionally, continuous monitoring of each critical control point is crucial for effective *Trichinella* spp. detection in infected meat [8].

In 2011, the relevant authorities of the Andalusia region in southern Spain developed a Quality Management System (QMS) based on ISO/IEC 17025 [6]. This QMS was applied to slaughterhouses and game handling establishments conducting *Trichinella* spp. testing without official accreditation but under the supervision of the relevant authority. The QMS not only enabled the implementation of high standards in *Trichinella* spp. analysis but also facilitated the identification and correction of practices that did not meet the minimum required standards.

Canonical Discriminant Analysis (CDA) has been proposed as a statistical alternative to tailor HACCP plans (Hazard Analysis and Critical Control Points) that are to be applied to other substances and products for human consumption like bottled water and allows for a comprehensive comparison of practices implemented across different facilities or brands. It provides valuable insights into the specific impact of combinations of factors, which can serve as pivotal points in discerning variations in practice application among these facilities or brands. Additionally, CDA facilitates the exploration of similarities and dissimilarities in the practices adopted by these facilities, unveiling clustering patterns that aid in effectively addressing potential issues [9]. To enhance the reliability of these statistical tools, CDA methods are often complemented by cross-validation techniques. This approach not only helps in pinpointing potential problems along the operational chain but also aids in characterizing the nature and associated risks of these issues [9]. CDA statistical tools are routinely followed by cross-validation techniques, which in turn can help identify issues along the chain and determine the potential nature of the issues and the risk that they imply.

Therefore, the objective of this study is to retrospectively analyze the outcomes of implementing the QMS in slaughterhouses and game handling establishments involved in *Trichinella* testing in southern Spain. This translatable discriminant tool permits us to assess the development of quality assurance practices across internal laboratories using DCA and the follow-up cross-validation methods. In turn, the outcomes of the present paper may provide an insight on which items or issues along the implementation of the QMS act as critical discriminant points across laboratories, thus assisting in the determination of the *Trichinella* risk that their occurrence and frequency may imply for the human food supply chain.

## 2. Materials and Methods

### 2.1. Study Units and Period: Satellite Laboratories in Southern Spain

The present study evaluates the implementation of a QMS based on UNE/EN ISO 17025 [6] in a total of 18 Satellite Laboratories (SLs), 7 of which were in the province of Cordoba (4 in slaughterhouses and 3 in game handling establishments) and 11 in the province of Seville (9 in slaughterhouses and 2 in game handling establishments). A description of the particular activity carried out in each SL can be found in Table 1.

These SLs were chosen upon a selection criterion for the LSs (Laboratory Services) based on the information provided, either quantitatively (number of audits conducted over a period of time) or qualitatively, which refers to the interest that the LS may have based on factors such as their activity (e.g., pig slaughter, game meat processing related to wild boars, or equine slaughter) or ownership (public or private). With this approach as a starting point, SLs were selected for which data from at least 4 internal audits were available from the period of 2012 to 2018.

The SLs, being facilities within slaughterhouses and game handling establishments, function as internal laboratories of the establishment, equipped with the necessary materials and resources to perform the analytical determination in question. Ensuring the reliability of the result is an essential requirement for accepting its validity, and according to EU regulatory requirements (Regulation EU 2015/1375 [5]), it must be supported by a QMS as a guarantee of that reliability.

The establishments opted for the alternative of tutelage, and under the supervision of the Official Veterinary Service (SVO), they initiated a process to adapt the facilities where the trichinellosis investigation was carried out and acquire the necessary materials and resources to adapt to a new approach to research on *Trichinella* spp. This was carried out under a quality assurance system based on the UNE/EN ISO 17025 standard [6] and under the supervision of an accredited laboratory, designated as the reference laboratory (Public Health Laboratory, PHL), which depended on the competent authority. The designed system relied on the necessary participation of the economic operator (EO), the SVO, and thePHL, with the collaboration of all three being essential for the effective and proper functioning of the QMS.

### 2.2. Study Sample

A total of 3023 obtained data points, relative to each deviation type and subtype, found for each Satellite Laboratory per year from 2012 to 2018 were considered.

### 2.3. Audits

The criteria considered were established in Regulation No. 2075/2005 of December 5th [10], which established specific rules for official controls (audits) for the presence of *Trichinella* in meat and its amendments, in force at the beginning of this initiative, and which was repealed by Regulation 2015/1375 [5].

A checklist of 36 questions or issues was used to obtain the basic information from each LS and assess their initial conditions before the implementation of the QMS.

The scope of the assessment consisted of evaluating aspects related to

Facilities requirements: premises where the investigation of *Trichinella* larvae is carried out.Technique requirements (assay): investigation of *Trichinella* larvae via hydrochloric-peptic digestion in fresh meat samples from controlling housing pigs (except for animals coming from a holding or a compartment officially recognized as applying controlled housing conditions in accordance with Annex IV—article 3, (EU) 2015/1375) [5] and wild boars, using the method of collective sample digestion with a magnetic stirrer (reference method).Activity data and records: existing records on the assay and its results, quality, and traceability.

For the Quality Management System (QMS) investigated here, exclusive documents (procedures) were developed, and others were designed for information recording purposes (formats), which had an individual coding but allowed for differentiation of each participating establishment. There was a necessary correlation between the aspects covered by the UNE/EN ISO 17025 standard [6] and the developed documentation.

The audits were carried out using a checklist of requirements and a questionnaire regarding the dependencies and technical requirements. Subsequently, a report on the audit was prepared, documenting the observed findings. The findings could be categorized as deviations (if they represented non-compliance or potential non-compliance in the future, for example), and findings that could contribute to strengthening the QMS and be considered improvement actions could also be identified. Deviations could be classified as Observations (OBS) or Non-Conformities (NC), with NCs having greater negative relevance in terms of the commitment to implementing the QMS. Both types required corrective actions, the effectiveness of which was subsequently verified, and all actions taken were documented.

The provided data inform about the situation of the LSs at the time of the audit and allow for an assessment of the degree of implementation and commitment to the QMS, as well as the reliability that should support the test result.

### 2.4. Deviation Classification and Coding

The classification of deviations was carried out to prioritize them based on importance (Observation–Non-Conformity), aspect (Technical Requirement–Management Requirement), and affected content (execution of the technique; equipment, material, and reagents; qualification; quality assurance; records, formats, and other documents and their control; and others). This classification aims to establish a system of grouping findings that facilitates their identification and possible significance regarding the reliability of the result, which is the main objective of the analysis, and coincides with Gajadhar, Noeckler, Boireau, Rossi, Scandrett, and Gamble [4], who affirm that the quality and accuracy of *Trichinella* spp. testing is dependent on the proper performance of the digestion method, the appropriate sample collection based on the target species, adequate facilities, equipment and consumables, accurate verification of findings, and proper documentation of the results. Thus, the goal is to identify findings as potential deviations and classify them based on their importance within the affected area.

Out of all the recorded deviations, a classification into two main groups was performed to create a set of criteria that helps classify the findings. These two groups are based on whether they affect technical aspects (related to the technical execution and its immediate environment of influence) or management and documentation aspects (related to records and other documents, whose existence is necessary but does not compromise the actual execution of the test).

The documented findings may be related to technical requirements or management requirements and may have different intensities, measured by the degree to which they affect the validity of the activity results (whether they question their validity or not), reveal serious non-compliance with management requirements, or occur in isolated or sporadic instances without affecting the activity results or questioning the consistency in the provision of activities. Depending on the degree of impact, these deviations from regulatory requirements can be classified as Non-Conformities or Observations.

The decision to consider a finding a Non-Conformity (NC) or Observation (OB), regardless of the affected scope (technical requirement or management requirement), depends on the significance or severity that the auditor believes it has in compromising the outcome.

After this initial classification, a classification was carried out based on the affected scope (management requirements/technical requirements), and within these scopes, different components can be affected. Six components were identified, which are linked to technique and/or test information; equipment, materials, reagents; qualification; quality assurance; records, forms, and other documents; and other components.

This categorization of components allows for the establishment of the same number of types of findings (6 types, identified by numbers 1 to 6), which are further broken down into subtypes (up to a maximum of 7, identified by letters from a to g. Type 3 does not have subtypes). The defined types include the following content, which provides guidance on the findings included within their scope.

This classification allows for the differentiation of 12 possible Non-Conformities (NCs) and 12 possible Observations (OBS), each with their respective subtypes, which reflect the degree of compliance with the QMS and the intensity or scope of the described deviation.

Grouping the findings into categories based on areas and types facilitates their classification, ensuring a regulated criterion and improving uniformity in the evaluation of findings.

This classification of deviations based on areas and types should be complemented by considering the importance of deviations from the perspective of issuing a valid and reliable result. It expands the dimension of the QMS beyond the analytical result itself, as not all types of findings equally affect the concepts of validity and reliability.

### 2.5. Canonical Discriminant Analysis

Canonical Discriminant Analyses (CDAs) were performed to design a tool that enables the classification of SLs while determining whether linear combinations of measures of quality-assurance-related traits describe within- and between-SL clustering patterns. The explanatory variables used for the present analyses were the essay number (number of essays seeking the detection of *Trichinella larvae*), positive levels (number of *Trichinella larvae* cases found), auditor combinations (random combinations of the three auditors performing the audits), deviation frequency description (description of the frequency of occurrence of each deviation, Figure 1), risk description (description of the risk depending on the punctuation obtained after each audit, Figure 2), and risk punctuation on a scale from insignificant (≤16) to imminent (≥65) (Figure 2). Additionally, the number of deviations classified per scope (either management or technical requirement), type (either Observation or Non-Conformity), and each subtype (a to f) were considered as well (Table 2). Each SL was considered on one level within the laboratory clustering criterion.

To create a territorial map that could be easily understood, we utilized canonical relationships based on traits to visualize the differences between groups. To select the most relevant variables, we employed regularized forward stepwise multinomial logistic regression algorithms. To ensure fairness and prevent the impact of varying sample sizes on the classification accuracy, we applied regularization to the priors based on the group sizes, considering the prior probability of commercial software (SPSS Version 26.0 for Windows, SPSS, Inc., Chicago, IL, USA). This approach aimed to avoid any bias caused by unequal group sizes and to enhance the overall quality of the classification results [11].

The same sample size contexts as those used in this study across groups have been reported to be robust. In this regard, some authors have reported a minimum sample size of at least 20 observations for every 4 or 5 predictors, and the maximum number of independent variables should be n−2, where n is the sample size, to palliate possible distortion effects [12,13].

Consequently, the present study used a 4- or 5-times higher ratio between observations and independent variables than those described above, which renders discriminant approaches efficient. Multicollinearity analysis was run to ensure independence and a strong linear relationship across predictors. Variables chosen by the forward or backward stepwise selection methods were the same. Finally, the progressive forward selection method was performed, since it requires less time than the backward selection method.

The discriminant routine of the Classify package of SPSS version 26.0 software and the canonical discriminant analysis routine of the Analyzing Data package of XSLTAT software (Addinsoft Pearson Edition 2014, Addinsoft, Paris, France) were used to perform canonical discriminant analysis.

#### 2.5.1. Multicollinearity Preliminary Testing

Multicollinearity refers to the linear relationship among two or more variables, which also means there is a lack of orthogonality among them. Multicollinearity analysis is crucial in improving the reliability of Hazard Analysis and Critical Control Point (HACCP) evaluations. It involves assessing the relationships between different variables or factors studied in HACCP. Detecting and quantifying multicollinearity helps in identifying highly correlated variables, which can lead to inaccurate or unstable results. By addressing multicollinearity, analysts can isolate the independent effects of each factor, resulting in a more accurate understanding of critical control points in the HACCP system. This, in turn, enhances the precision of risk assessments and strategies for ensuring food safety and quality. Different methods are available to detect multicollinearity, and the most widely used are variance inflation factor (VIF) and tolerance [14]. VIF is a ratio of variance in a regression model with multiple attributes divided by the variance of a model with only one attribute [15]. Explained more technically and exactly, multicollinearity occurs when *k* vectors lie in a subspace of dimension less than *k*. Multicollinearity can explain a data-poor condition, which is frequently found in observational studies in which the researchers do not interfere with the study. Thus, many investigators often confuse multicollinearity with correlation. Whereas correlation is the linear relationship between just two variables, multicollinearity can exist between two variables or between one variable and the linear combination of the others. Therefore, correlation is considered a special case of multicollinearity. A high correlation implies multicollinearity, but not the other way around. Before performing the statistical analyses per se, a multicollinearity analysis was run to discard potential strong linear relationships across explanatory variables and ensure data independence. In this way, before data manipulation, redundancy problems can be detected, which limits the effects of data noise and reduces the error term of discriminant models. The multicollinearity preliminary test helps identify unnecessary variables which should be excluded, preventing the overinflation of variance explanatory potential and type II error increase [16]. The variance inflation factor (VIF) was used to determine the occurrence of multicollinearity issues. The literature reports a recommended maximum VIF value of 5 [17]. On the other hand, tolerance (1 − R^2^) concerns the amount of variability in a certain independent variable that is not explained by the rest of the dependent variables considered (tolerance > 0.20) [18]. The multicollinearity statistics routine of the describing data package of XSLTAT software (Addinsoft Pearson Edition 2021, Addinsoft, Paris, France) was used. The following formula was used to calculate the VIF:VIF = 1/(1 − R^2^),(1)
where R^2^ is the coefficient of determination of the regression equation.

In the present study, four rounds were needed to rule out all the potential factors involved in the occurrence of problems of multicollinearity, discarding one of the factors at each round.

#### 2.5.2. Canonical Correlation Dimension Determination

The maximum number of canonical correlations between two sets of variables is the number of variables in the smaller set. The first canonical correlation usually explains most of the relationships between different sets. In any case, attention should be given to all canonical correlations, despite reporting of only the first dimension being common in previous research. When canonical correlation values are 0.30 or higher, they correspond to approximately 10% of the variance explained.

#### 2.5.3. Canonical Discriminant Analysis Efficiency

Wilks’ lambda test evaluates which variables may significantly contribute to the discriminant function. When Wilks’ lambda approximates 0, the contribution of that variable to the discriminant function increases. χ^2^ tests the Wilks’ lambda significance. If significance is below 0.05, the function can be concluded to explain the group adscription well [19].

#### 2.5.4. Canonical Discriminant Analysis Model Reliability

Pillai’s trace criterion, as the only acceptable test to be used in cases of unequal sample sizes, was used to test the assumption of equal covariance matrices in the discriminant function analysis [20]. Pillai’s trace criterion was computed as a subroutine of the Canonical Discriminant Analysis routine of the Analyzing Data package of XSLTAT software (Addinsoft Pearson Edition 2014, Addinsoft, Paris, France). A significance of ≤0.05 is indicative of the set of predictors considered in the discriminant model being statistically significant. Pillai’s trace criterion is argued to be the most robust statistic for general protection against departures from the multivariate residuals’ normality and homogeneity of variance. The higher the observed value for Pillai’s trace is, the stronger the evidence is that the set of predictors has a statistically significant effect on the values of the response variable. That is, the Pillai trace criterion shows potential linear differences in the combined quality-assurance-related traits across SL clustering groups [21].

#### 2.5.5. Canonical Coefficients and Loading Interpretation and Spatial Representation

When CDA is implemented, a preliminary principal component analysis is used to reduce the overall variables into a few meaningful variables that contributed most to the variations between SLs. The use of the CDA determined the percentage assignment of SLs within its own group. Variables with a discriminant loading of ≥|0.40| were considered substantive, indicating substantive discriminating variables. Using the stepwise procedure technique, non-significant variables were prevented from entering the function. Coefficients with large absolute values correspond to variables with greater discriminating ability.

Canonical loadings represent the relationships between the original variables and the canonical variables derived from the data. They help identify which original variables contribute the most to the canonical correlation between datasets, shedding light on the most influential factors in content personalization. Canonical coefficients, on the other hand, provide a numerical representation of the strength and direction of these relationships, offering valuable insights into how specific variables impact content creation. By analyzing canonical loadings and coefficients in the HAPCC model, content creators and data scientists can optimize the personalization process, ensuring that content is finely tuned to meet the diverse needs and preferences of users.

Data were standardized following procedures reported by Manly and Alberto [22]. Then, squared Mahalanobis distances and principal component analysis were computed using the following formula:$$D_{\mathrm{ij}}^{2}={(Ῡ}_{i}-{Ῡ}_{j}){\mathrm{COV}}^{-1}{(Ῡ}_{i}-{Ῡ}_{j}),$$
where $D_{ij}^{2}$ is the distance between population i and j; COV^−1^ is the inverse of the covariance matrix of measured variable x; ${Ῡ}_{i}$ and ${Ῡ}_{j}$ are the means of variable x in the ith and jth populations, respectively.

In our study, the results were spatially represented through the creation of a territorial map, which served as a visual representation of the geographical distribution of key variables. The squared Mahalanobis distance matrix was converted into a Euclidean distance matrix, and a dendrogram was built using the underweighted pair-group method arithmetic averages (UPGMA; Rovira i Virgili University, Tarragona, Spain) and the Phylogeny procedure of MEGA X 10.0.5 (Institute of Molecular Evolutionary Genetics, The Pennsylvania State University, State College, PA, USA).

The practical implications of the classification accuracy and Press’ Q statistic are profound. A high classification accuracy signifies the reliability of our model in assigning each audit to the appropriate satellite laboratory, which is crucial in decision making processes. In contrast, a strong Press’ Q statistic indicates that the model is capable of accurately predicting new, unseen data, thus enhancing its real-world applicability. These statistical metrics are invaluable for policy makers, urban planners, and other stakeholders, as they inform data-driven decisions that can have a lasting impact on territorial management and resource allocation. It is important to note that all relevant references to regulations, standards, and best practices were meticulously cited throughout the study to ensure the credibility and replicability of our findings.

#### 2.5.6. Discriminant Function Cross-Validation

To assess the accuracy of our discriminant functions, we employed a rigorous cross-validation procedure. This involved splitting the dataset into training and testing subsets, ensuring that the model’s performance was evaluated on unseen data. Cross-validation helps mitigate overfitting and provides a robust estimation of the model’s predictive power. Furthermore, we calculated Press’ Q statistic, which measures the sum of squared prediction errors and provides a valuable assessment of model fit. High Q values indicate excellent model performance.

Afterwards, to determine the probability that an audit of a SL of an unknown background belongs to a particular SL [23], the hit ratio parameter was computed. For this, the relative distance of each particular audit to the centroid of its closest SL was used. The hit ratio is the percentage of correctly classified audits that are correctly ascribed to the SLin which they were performed. The leave-one-out cross-validation procedure is used as a form of significance to consider if the discriminant functions can be validated. Classification accuracy is achieved when the classification rate is at least 25% higher than that obtained by chance.

Press’ Q statistic can support these results, since it can be used to compare the discriminating power of the cross-validated function, as follows:$$Press^{\prime }Q=\frac{[{n-(n\prime K)]}^{2}}{n\left(K-1\right)},$$
where n is the number of observations in the sample; n’ is the number of observations correctly classified; and K is the number of groups.

The value of Press’ Q statistic must be compared with the critical value of 6.63 for χ^2^ with a degree of freedom at a significance of 0.01. When Press’ Q exceeds the critical value of χ^2^ = 6.63, the cross-validated classification can be regarded as significantly better than chance.

## 3. Results

### 3.1. Multicollinearity Prevention: Preliminary Testing

A summary of values for VIF and tolerance is reported in Appendix A. Variables whose VIF values were ≥5 were discarded from further analyses. Thus, all traits were removed for the following statistical analyses except for auditor combination and risk description.

### 3.2. Canonical Discriminant Analysis

#### 3.2.1. Canonical Discriminant Analysis Model Reliability

A significant Pillai’s trace criterion determined that discriminant canonical analysis was feasible (Pillai’s trace criterion = 0.9830; F (Observed value) = 17.3469; F (Critical value) = 1.1766; df1 = 187; df2 = 33,055; *p*-value < 0.0001). As reported in Table 3, seven out of the eleven discriminant functions designed after the analyses presented a significant discriminant ability. The discriminatory power of the F1 function was high (eigenvalue of 0.5356; Figure 2), with ≈64% of the variance being explained by F1 and F2.

#### 3.2.2. Canonical Coefficients, Loading Interpretation, and Spatial Representation

Variables were ranked depending on their discriminating properties. For this, a test of equality of group means across SLs was used (Table 4). Lower values of Wilks’ lambda and greater values of F indicate a better discriminating power, which translates into a better position in the rank. The analyses revealed that either auditor combination or risk description significantly contributes (*p* < 0.05) to the discriminant ability of significant discriminant functions.

Standardized discriminant coefficients measure the relative weight of auditor combinations and risk description in the discriminant functions (Figure 3).

Out of the seven significant discriminant functions (Table 3), only the two most relevant functions were used to build a standardized discriminant coefficient biplot, capturing the highest fraction of variance (Figure 3). In this regard, those variables whose vector extends further apart from the origin most relevantly contributed to the first (F1) and second (F2) discriminant functions.

Table 5 suggests a clear differentiation among the SLs considered in the analyses. The relative position of centroids was determined through the substitution of the mean value for observations in each term of the first two discriminant functions (F1 and F2). The larger the distance between centroids, the better the predictive power of the canonical discriminant function in classifying observations. Appendix A reports the results obtained in the classification and leave-one-out cross-validation. A Press’ Q value of 294.299 (N = 3023; n = 384; K = 18) was obtained. Therefore, it can be considered that predictions were significantly better than chance at 95% [24].

Additionally, to evaluate the proximity between SLs, Mahalanobis distances were represented (Figure 4). Two main clusters are formed, the first represented by SL1, which was the most distant SL from the rest (Mahalanobis distance of 2.8820) when auditor combinations and risk description levels are considered, and the second subcluster comprising the seventeen remaining SLs (1.7180). A progressive segregation of SLs occurs within the second cluster into two closer subclusters. The first, at 0.4640 and comprising SL 7, 8, 9, 20, 25, 27, and 28, and a second one comprising the rest of the SLs at 0.2680. It is within the second subcluster that two sub-subclusters are formed. The first sub-subcluster, comprising SL 13, 19, 21, and 22 at 0.1460, accounts for two additional branches at 0.7500, and the second sub-subcluster (0.4380) comprises the rest of the SLs.

## 4. Discussion

The previous results analyze the recommendations of the International Commission on Trichinellosis (ICT) at the time in terms of facilitating reliable test results when laboratories operate within a QMS and set the critical aspects of the analytical process, in this case when satellite laboratories are supervised by the official authority audits and do not work under their own accreditation. Also, the performances of different combinations of auditors reveal the extreme importance of training and qualification on audit guidelines.

The discussed findings and recommendations are highly relevant to the broader context of meat inspection, *Trichinella* spp. control, and quality assurance in satellite laboratories (SLs). They emphasize the critical role of Quality Management Systems (QMS) in ensuring the reliability of test results, which is pivotal for public health, surveillance, and trade in the meat industry. The research underscores that when satellite laboratories are supervised by official authority audits and do not operate under their accreditation, there is a pressing need to focus on technique procedures and adherence to established standards, as these aspects significantly impact the accuracy of *Trichinella* spp. testing.

The study acknowledges several limitations, such as variations in resource allocation and pressures on SGC activities between different satellite laboratories, which can influence the effectiveness of quality assurance. Additionally, the research highlights the potential subjectivity in audit outcomes when conducted by a single auditor, emphasizing the importance of conducting audits simultaneously by at least two auditors for more consistent and reliable results.

Practical recommendations based on the findings include implementing a training system for auditors to ensure uniformity in audit procedures and compliance with QMS standards. The study also suggests that internal on-site audits by the relevant authority should be conducted alongside internal audits by the testing laboratory and third-party external audits to enhance oversight and quality assurance.

In summary, this study’s contribution to the field of meat inspection and laboratory quality assurance is significant, as it identifies key factors affecting the reliability of *Trichinella* spp. testing and offers practical solutions to address these issues. The emphasis on adherence to QMS standards, consistent auditing practices, and the importance of collaboration between different stakeholders in the meat industry underscores the importance of maintaining high-quality testing and inspection procedures to safeguard public health and trade standards.

Considering the classification of the obtained “findings” for all analyzed aspects, all types (except Type 3) were distinguished by subtype a (either referring to Observations or Non-Conformities) with the highest detection frequency. For instance, with respect to aspects affecting the technique/information of the trial (Type 1), we observed that the most frequent finding during audits was regarding the following aspects: “The reliability of the result may be questioned by performing the necessary test or calculations incorrectly or without following the instructions indicated. Failure to follow the current Technical Procedure of the test. Use material and/or equipment that is not supported, uncontrolled or deficient”. These aspects substantiate reliable test results that are essential for public health, surveillance, and trade. [25] concludes that test results are reliable when laboratories operate within a valid quality assurance (QA) program, which includes a validated test method, procedures to confirm laboratory capability, and protocols for documentation, reporting, and monitoring. When any of these aspects are more frequently detected, it is also expected that some other related practices (included within subtype “b” of the same type) also occur (Appendix A). In other words: when a failure to follow the current technical procedure of the test is the most frequent finding (Type 1, subtype a), it is also expected that some other deviations (either Observations or Non-Conformities), such as the record of the correct temperature or digestion % of the samples during the procedure (Type 1, subtype b), also occurs. The above reasons establish Type 1 (subtype a) as a Critical Control Point in the QMS.

For practices performed in SLs affecting the technique (Type 1), our results are in accordance with previous studies [26] establishing that the ISO Standard provides the basic principles and properties of the essential steps of the method and highlights the Critical cControl Points (CCPs) of the procedure. In particular, the importance of the correct sequence of mixing of the digest fluid (CCP), as well as the reading of the sediment, is highlighted, since repeated clarification steps (as indicated in the Regulation) can lead to larval loss, while unclear sediment can affect the readability of the digest fluid (CCP). Similarly, Marucci et al. [27] conclude that the heterogeneity of the performance of National Reference Laboratories could be related to both the technical expertise of the individuals performing the test and the equipment used for the test. The same authors [27] report that no significant association (*p* = 0.06) was found between the test sensitivity and the number of digestions performed. As previously indicated in the manuscript in terms of risk punctuation, it is worth mentioning that regardless of the technical method during the control of *Trichinella* spp., the greater the number of tests performed, the greater the probability of positive cases that can be detected.

Overall, based on the collected information, it was found that the major deficiencies were related to technique procedures and the need to implement actions to ensure the reliability of the results from the perspective of the UNE/EN ISO 17025 standard [6], rather than regarding the facilities or the execution of the assay itself. It is worth noting the effort, willingness, and professional commitment of the personnel performing the assay. Some LSs made attempts to address the deficiencies, either in terms of records or other documents, or in terms of temperature control or other verifications, with actions intended to provide reliability guarantees. However, these actions were considered insufficient to support the reliability of the results. It should be considered that designing, implementing, and maintaining a QMS like the one mentioned required dealing with a set of documentation (instructional procedures, reports, records) and actions (calibrations, verifications, maintenance, training, supplier evaluations, quality assurance, etc.) that had not typically been suggested for this type of facilities. Therefore, the need in this regard and the extensive documentation required explained the findings and justified the corrective measures that needed to be taken to address the deviations found.

To ensure that all requirements of good testing practices are met, a formal system of scheduled on-site audits of the testing laboratory is required in order to identify and document deficiencies in the QMS for testing, specify corrective actions, and achieve a satisfactory resolution [7]. Then, a standardized audit checklist should be used to ensure that all aspects of the QMS for *Trichinella* spp. testing are reviewed. Key elements should include facilities, personnel, training, equipment, quality manual (or similar), standard operating procedures (SOPs), and record keeping. In this sense, audit results may be used to evaluate if laboratory practices follow established procedures for *Trichinella* spp. inspection [28] and to assess the audit procedure itself. In this work, we have observed that the combination of auditors plays a key role in/shows a major influence on the overall outcome of each SL assessment in terms of detected practices that are not in compliance with the QMS. Based on the data shown in Appendix A, we detected that when the audit is carried out by a single auditor, the difference between SLs in terms of detected deviations increases, in contrast to what was observed when the audit was performed simultaneously by two auditors. This was detected in all situations, regardless of the combination of auditors/the auditors’ partnership, with the exception of auditor C: when auditor C audited alone, no major differences were detected between the assessment of SLs. We hypothesize that when two auditors audit together, a consensus can be reached between both, causing no major differences to be detected between SLs, and thus, probably reducing the degree of subjectivity during audits.

In *Trichinella* spp. control, quality assurance requires the use of reliable methods [28], not only to be applied at slaughterhouses and game meat handling establishments, but also alongside audits. Therefore, we suggest that it is highly recommended that on-site audits carried out by the relevant authority are performed simultaneously at least by two auditors, and that an auditor training system should be implemented/is to be implemented by the relevant public health authority, with the aim of guaranteeing/assuring that audits are conducted homogeneously among auditors and in compliance with the QMS on *Trichinella* spp. control. A recent study based on data collected from 62 clinical audits also highlighted staff training in terms of performing audits [29]. In this regard, the assurance of adequate compliance and oversight lies with the relevant veterinary authority [28]. Furthermore, it has been suggested that internal on-site audits performed by the relevant authority should be conducted together with internal audits performed by the testing laboratory and with an external on-site audit carried out by a third party, which may be the pertinent national accrediting body for ISO/IEC 17025-accredited testing laboratories [7]. Additionally, although audits should usually be conducted at least once annually [4], OIE states that the frequency of audits should occur based on risk, considering the *Trichinella* spp. testing laboratory result [28].

Figure 4 suggests that establishments LS1 and LS4, notwithstanding their shared involvement in the porcine slaughtering industry, manifest distinctions in their operational paradigms and activity magnitude.

LS1 represents an aged facility experiencing a gradual decline in activity levels. Within LS1, the establishment and execution of the Quality Management System (SGC) rested solely upon the shoulders of the Veterinary Officer (SVO). The SVO’s responsibilities were distributed among multiple individuals on a rotating basis, thereby encompassing the full spectrum of SGC duties and extending beyond the scope of their official role. Additionally, the limited number of pig slaughters conducted at LS1 had a detrimental effect on the risk assessment pertaining to the detection of Trichinella larvae, as evidenced by the results of 475 tests.

Conversely, LS4 operated as a high-capacity slaughterhouse, wherein the administration of the SGC was undertaken collaboratively by LS4 itself and the SVO. In the context of LS4, the SVO focused exclusively on official duties. The substantial volume of slaughter conducted at LS4 exerted a commensurate influence on the associated risk, resulting in outcomes that are congruent with those of LS1, as indicated by 1491 tests. This situation engenders a scenario marked by variances in resource allocation and the pressures exerted on SGC activities and management across the two establishments.

In both establishments, the auditing process was conducted by three auditors. However, one auditor conducted a maximum of two audits, whereas the remaining two auditors executed between four and five audits each. Notably, there were no statistically significant disparities in the number of audits performed between these two auditors. Specifically, one auditor conducted five site visits to LS1 and four to LS4, while the other auditor completed four visits to LS1 and an equal number to LS4. Additionally, both auditors conducted joint audits once at LS1 and twice at LS4. These empirical findings suggest that audit results were subjected to reasonably uniform interpretive criteria, with the discernible discrepancies between LS1 and LS4 primarily attributable to their distinct operational realities.

## 5. Conclusions

Audit findings primarily target subtype a issues in Type 1, encompassing incorrect procedures, non-compliance with technical guidelines, and equipment inadequacies, underscoring the importance of reliable test results for public health and trade. Frequent subtype deviations often coincide with subtype b issues, emphasizing Type 1 as a Critical Control Point in the QMS. Adherence to ISO standards, focusing on critical control points like proper digest fluid mixing and sediment reading, is vital for slaughterhouse quality. The performance of National Reference Laboratories depends on personnel expertise, equipment quality, and test volume, impacting positive case detection. This study highlights deficiencies in technique procedures, necessitating measures for result reliability, due to facilities’ unfamiliarity with extensive QMS documentation. Our recommendations include dual-auditor on-site audits for consistency and reduced subjectivity, auditor training, and collaborative efforts to enhance oversight and compliance. The operational disparities between LS1 and LS4 result from their unique contexts.

## Figures and Tables

**Figure 1 foods-12-04186-f001:**
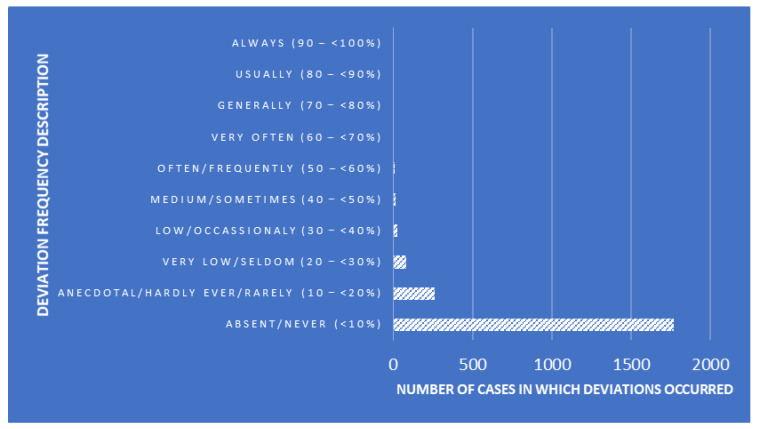
Deviation frequency description and number of cases in which deviations occurred.

**Figure 2 foods-12-04186-f002:**
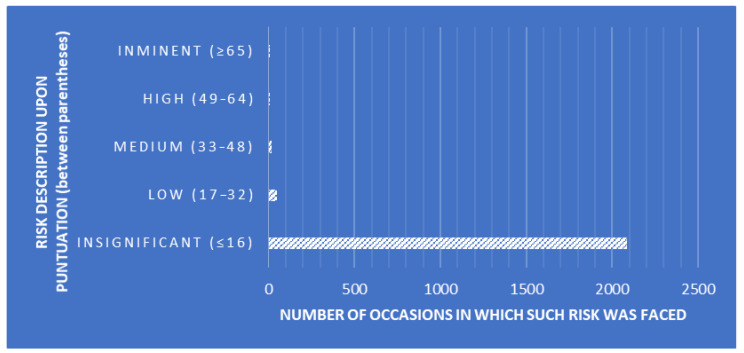
Risk description upon punctuation (between parentheses) and number of occasions in which such risk was faced.

**Figure 3 foods-12-04186-f003:**
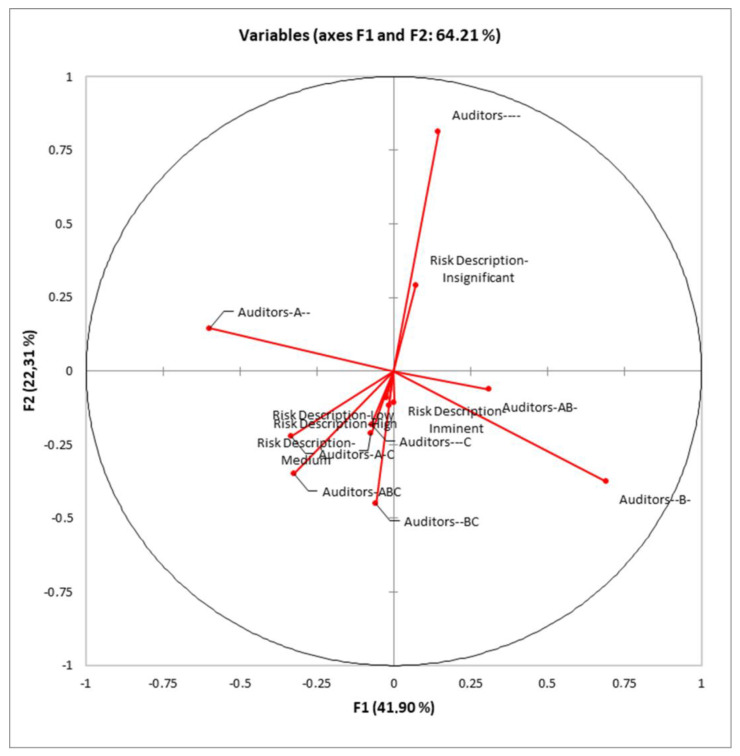
Standardized canonical discriminant function coefficients for each auditor combination and risk description level.

**Figure 4 foods-12-04186-f004:**
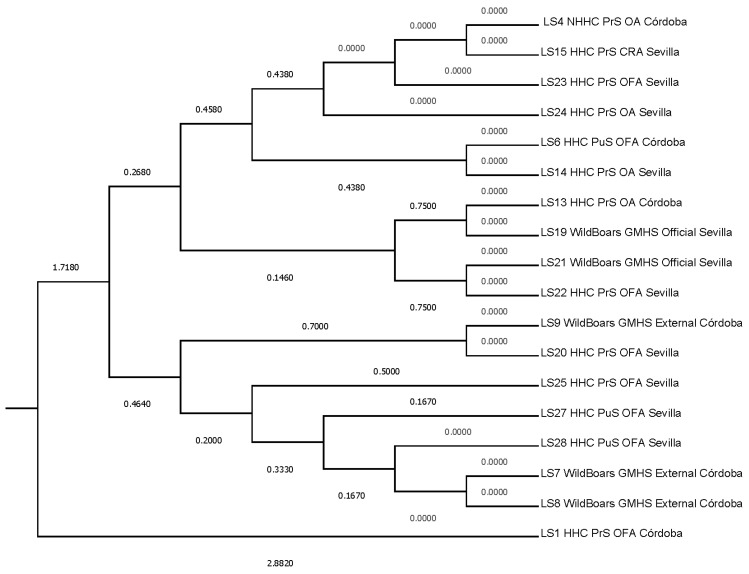
Tree representing Mahalanobis distances across SLs. Classification was as follows; Breed (HHC: Handling Housing Conditions vs. NHHC: Non-Handling Housing Conditions), Establishment Type (PrS: Private Slaughterhouse, PuS: Public Slaughterhouse, and GMHS: Game Meat Handling Establishment), Audit Nature (OA: Own Audit, CRA: Client-requested Audit, and OFA: Official Audit.

**Table 1 foods-12-04186-t001:** Description of the particular activity carried out by each satellite laboratory.

Satellite Laboratory	Activity
SL 1	Slaughterhouse ^1^
SL 4	Slaughterhouse
SL 6	Slaughterhouse
SL 7	Game handling establishment ^2^
SL 8	Game handling establishment
SL 9	Game handling establishment
SL 13	Slaughterhouse
SL 14	Slaughterhouse
SL 15	Slaughterhouse
SL 19	Game handling establishment
SL 20	Slaughterhouse
SL 21	Game handling establishment
SL 22	Slaughterhouse
SL 23	Slaughterhouse
SL 24	Slaughterhouse
SL 25	Slaughterhouse
SL 27	Slaughterhouse
SL 28	Slaughterhouse

The aforementioned establishments had the option to either implement UNE/EN ISO 17025, obtain accreditation, and thus not be subject to the oversight of the competent authority, or choose the mentioned oversight, where the issuance of a compliant or non-compliant result must be validated by the official veterinary service (OVS) responsible for it. ^1^ Pig slaughter; ^2^ Handling of wild boar.

**Table 2 foods-12-04186-t002:** Classification of deviations by type and subtype according to findings. Examples for each type of deviation are given in Appendix A as Appendix A.

Classification of Findings Affecting QMS Requirements	
DEVIATIONS (According to severity of the finding)	Non-Conformities (NC)
	Observations (OB)
AMBIT (According to affected requirement)	Technical Requirements (TR)
	Management Requirements (MR)
TYPES (Depending on affected component)	SUBTYPES
TYPE 1 Affects TECHNIQUE and/or trial information.	SUBTYPE (a) The reliability of the result may be questioned by performing the necessary test or calculations incorrectly or without following the instructions indicated. Failure to follow the current Technical Procedure of the test. Use of material and/or equipment that is not supported, uncontrolled or deficient.
	SUBTYPE (b) Failure to record in the test report data that could only be known if they are collected in the test report (i.e., temperature, sieve weight, digestion %, identification of samples, reagents and/or equipment if more than one is in use). Inappropriate use of registry spaces. Untraceable information exists.
TYPE 2 Affects EQUIPMENT, MATERIAL, REAGENTS	SUBTYPE (a) There is no Maintenance and Calibration Plan (MANCA Plan)
	SUBTYPE (b) They affect equipment and/or consumables in their identification or registration, technical characteristics, use, operation, control, high or low, without compromising the reliability of the result.
	SUBTYPE (c) Some consumable equipment/material is missing.
	SUBTYPE (d) Equipment instructions are missing or not located/provided.
	SUBTYPE (e) There is equipment/material in poor condition.
	SUBTYPE (f) There is a lack of reagents for testing and/or for the disposal of positive samples with no alternative. The reagents are poorly controlled.
TYPE 3 Affects QUALIFICATION	NO SUBTYPES. They compromise the evidence of the qualification or training of the person(s) involved in the performance of the technique and/or the interpretation of the result.
TYPE 4 Affects QUALITY ASSURANCE	SUBTYPE (a) They affect internal quality controls (ICCs) or external quality controls.
	SUBTYPE (b) They affect the Corrective Action Plan (PAC) and/or the management of deviations.
	SUBTYPE (c) They concern calibrations, verifications, maintenance, and/or calibration labels, including the control of information or metrological traceability.
	SUBTYPE (d) They affect the quality certificates of the equipment or technical sheets of the reagents.
	SUBTYPE (e) They affect the ISO certificates of suppliers.
	SUBTYPE (f) They affect the record of deviations.
	SUBTYPE (g) They affect the conservation of reagents.
TYPE 5 Affects RECORDS, FORMATS, AND OTHER DOCUMENTS	SUBTYPE (a) Unfilled forms. Data are missing, erroneous, or illegible. There is a different signature than the one indicated. Data are missing from the test report not included in the TYPE 1 deviation.
	SUBTYPE (b) Missing, not locating/contributing, or not using current data record formats (FLS). Lack of use of the current primary data logging format.
	SUBTYPE (c) Missing, not located/provided, or using technical instructions (ITLS) in force.
	SUBTYPE (d) The technical procedure (PTLS) in force and/or the applicable regulations (EU Regulation) are missing, not located or provided, but it does not imply that the test is carried out incorrectly.
	SUBTYPE (e) There are deletions or unvalidated corrections. The space of a section in a format is used abusively. Ellipsis and/or quotation marks are used.
	SUBTYPE (f) No control is maintained of current copies of documents and/or obsolete documents or associated records/documents.
TYPE 6 Affects OTHER components	SUBTYPE (a) They affect the facilities (availability of hot/cold water, air conditioning, location, furniture, cleaning, or access).
	SUBTYPE (b) Improvement options are evident (e.g., in the identification of samples, in the management of information, use of better-quality consumables, completion, and custody of formats)
	SUBTYPE (c) They affect the elimination of positive samples (inadequate or non-existent containers, lack of reagents or system for it).
	SUBTYPE (d) The location of documents or files is not adequate or unknown. There are only equipment instructions in a language that is not Spanish and is unknown to the user.

**Table 3 foods-12-04186-t003:** Eigenvalues and cumulative variability explanatory power of the eleven discriminant functions revealed through CDA.

	Eigenvalue	Discrimination (%)	Cumulative %	Bartlett’s Statistic	*p*-Value
F1	0.5356	41.9027	41.9027	3349.7551	0.001
F2	0.2851	22.3057	64.2084	2059.7344	0.001
F3	0.1641	12.8383	77.0467	1305.2909	0.001
F4	0.1162	9.0873	86.1340	848.2920	0.001
F5	0.0847	6.6227	92.7567	517.7911	0.001
F6	0.0500	3.9118	96.6685	273.3983	0.001
F7	0.0303	2.3667	99.0352	126.6540	0.001
F8	0.0051	0.3990	99.4342	37.0188	0.605
F9	0.0038	0.2998	99.7339	21.7190	0.752
F10	0.0026	0.2004	99.9344	10.2175	0.855
F11	0.0008	0.0656	100.0000	2.5222	0.925

**Table 4 foods-12-04186-t004:** Eigenvalues and cumulative variability explanatory power of the eleven discriminant functions revealed through CDA.

Variable and Levels	Lambda	F	DF1	DF2	*p*-Value	Rank
Auditor B	0.77506781	51.30	17	3005	<0.0001	1
Auditor A	0.82025052	38.74	17	3005	<0.0001	2
No Auditor	0.82233103	38.19	17	3005	<0.0001	3
Auditors ABC	0.86314001	28.03	17	3005	<0.0001	4
Auditor C	0.89530283	20.67	17	3005	<0.0001	5
Auditors BC	0.90745129	18.03	17	3005	<0.0001	6
Auditors AB	0.90758069	18.00	17	3005	<0.0001	7
Risk Description-Insignificant	0.97437082	4.65	17	3005	<0.0001	8
Risk Description-Inminent	0.99205962	1.41	17	3005	0.11900639	-
Risk Description-High	0.99254927	1.33	17	3005	0.16521084	-
Risk Description-Low	0.99424836	1.02	17	3005	0.42913507	-
Auditors AC			17	3005		-
Risk Description-Medium			17	3005		-

**Table 5 foods-12-04186-t005:** Functions at the centroids for each SL across the main discriminant functions (F1 and F2).

	F1	F2
LS1	−1.5638	−0.9817
LS13	−0.9101	0.3640
LS14	−0.1990	−0.4345
LS15	0.5231	0.0584
LS19	0.3335	−0.0178
LS20	1.3449	−0.1281
LS21	−0.1049	0.3933
LS22	−0.3823	−0.0923
LS23	−0.5063	−0.3072
LS24	0.3813	0.2272
LS25	1.0697	−0.4016
LS27	−0.2419	1.1385
LS28	−0.0553	1.0039
LS4	−0.8039	−0.2707
LS6	−0.7195	0.2408
LS7	0.7550	−0.9245
LS8	0.4435	0.1898
LS9	0.6268	−0.0634

## Data Availability

The data that support the findings of this study are available from the corresponding author upon reasonable request.

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
