# Peer review of "A Quality Assurance Discrimination Tool for the Evaluation of Satellite Laboratory Practice Excellence in the Context of the European Official Meat Inspection for Trichinella spp."

_foods, 2023, doi:10.3390/foods12224186_

Round 1

Reviewer 1 Report

Comments and Suggestions for Authors

The article entitled "Quality Assurance Discrimination Tool for the Evaluation of Satellite Laboratory Practice Excellence in the Context of the European Official Meat Inspection for Trichinella spp." appears comprehensive in its scope and objectives, here are some points of improvement:

Start the abstract with a clearer and concise introduction about the importance of detecting Trichinella spp. in meat. There's a jump from QMS being applied in slaughterhouses to Canonical discriminant analyses (CDAs) without any bridge information, which can be confusing. Ensure that abbreviations are introduced before being used, for instance, QMS and SLs. "technical procedure issues were the most determinant risk increasing findings during audits" sounds awkward. Consider: "Technical procedure issues were the primary risk factors identified during audits."

Some sentences are repetitive with the abstract in the introduction. Consider rephrasing or presenting the information differently to maintain reader engagement. After explaining the importance of Trichinella spp. detection, move more fluidly into the importance of quality assurance in laboratories, making the transition smoother for readers. Give a brief overview of previous studies or existing systems before diving into specifics. This will help to contextualize the study. Expand on why the magnetic stirrer artificial digestion technique is considered the gold standard method. What makes it superior to other methods? When mentioning the EU data of 117 human cases, provide more context. Over what period were these cases reported? This could provide clarity on the current state of the problem. While mentioning various EU regulations and directives, provide a brief explanation or importance of each. Expand on what is meant by "in other substrates of human interest," this is ambiguous and might confuse readers unfamiliar with the topic.

The methodology section could benefit from improved organization and clarity. Consider breaking down the section into subsections or using bullet points to make it easier for readers to follow the steps of your research. Provide a more detailed explanation of what Satellite Laboratories (SLs) are and the Quality Management System (QMS) based on UNE/EN ISO 17025. This would help readers who may not be familiar with these terms to understand the context better. When discussing the selection criteria for the SLs, explain in more detail how the quantitative and qualitative factors were weighted or evaluated to select the laboratories. This will provide transparency in the selection process. The table describing the activities of each satellite laboratory (Table 1) could be formatted more clearly. Consider using a table format that includes headings for "Satellite Laboratory" and "Activity" to make it easier to read. In the section on "Deviation classification and Coding," provide examples or explanations for each type and subtype of deviation (e.g., Type 1, Subtype (a)). This will help readers understand the significance of each deviation type. Provide more context and explanation for the statistical analyses performed, such as Canonical Discriminant Analysis (CDA). Explain why CDA was chosen and how it relates to the research objectives. When discussing multicollinearity, explain the potential impact of multicollinearity on the results and why it is important to address it. Mention any actions taken to mitigate multicollinearity. In the section on canonical coefficients and loading interpretation, provides more information on how these coefficients were interpreted and their relevance to the study. Explain how the results were spatially represented, especially when discussing the creation of a territorial map. Provide details on the methodology used for this representation. Elaborate on the cross-validation procedure and its significance in assessing the accuracy of the discriminant functions. Explain the practical implications of the classification accuracy and Press' Q statistic. Ensure that all relevant references are cited, especially when referring to regulations or standards (e.g., Regulation EU 2015/1375).

Ensure that the presentation of results is clear and well-structured. Use subheadings, bullet points, or tables to organize the information for easier comprehension. Provide a brief explanation of the statistical tests used and their relevance to the analysis. This will help readers understand the significance of the reported values (e.g., VIF, F-statistic). While you've reported numerical results, add some interpretation to help readers understand the practical implications of the findings. What do these results mean in the context of the study's objectives? Consider adding visual aids (e.g., graphs, charts) to illustrate key findings, especially if the data is complex. Visual representation can make the results more accessible.

Start the discussion section by summarizing the key findings from the results section. Provide a concise overview of what the data revealed. Discuss how your findings align or differ from previous research in the same field. Highlight any new insights or contributions your study brings to the existing knowledge. Clearly state the implications of your findings. How do they relate to the broader context of meat inspection, Trichinella spp. control, and quality assurance in satellite laboratories? Acknowledge the limitations of your study. What constraints or challenges did you encounter during the research, and how might they have influenced the results? Offer practical recommendations based on your findings. What steps can be taken to address the issues identified in the satellite laboratories? How can the discrimination tool be improved or utilized effectively? Summarize the main points of the discussion and reiterate the significance of your study's contribution to the field of meat inspection and laboratory quality assurance.

Ensure consistent use of terms. For example, decide whether to use "Discriminant Canonical Analysis" or "Canonical Discriminant Analysis." While they aren't presented here, ensure that any figures or tables in the article are well-integrated and referenced in the text. Ensure that all claims or data points have appropriate and recent citations. There are instances where sentences can be shortened or made more concise, which would improve readability.

Remember, constructive feedback should be taken in the spirit of improving the quality and readability of the article. These suggestions are to aid in that process.

Comments on the Quality of English Language

Minor editing of the English language required

Author Response

Reviewer(s)' Comments to Author:

Reviewer 1

Comments and Suggestions for Authors

The article entitled "Quality Assurance Discrimination Tool for the Evaluation of Satellite Laboratory Practice Excellence in the Context of the European Official Meat Inspection for Trichinella spp." appears comprehensive in its scope and objectives, here are some points of improvement:

Start the abstract with a clearer and concise introduction about the importance of detecting Trichinella spp. in meat. There's a jump from QMS being applied in slaughterhouses to Canonical discriminant analyses (CDAs) without any bridge information, which can be confusing. Ensure that abbreviations are introduced before being used, for instance, QMS and SLs. "technical procedure issues were the most determinant risk increasing findings during audits" sounds awkward. Consider: "Technical procedure issues were the primary risk factors identified during audits."

Response: We followed reviewer’s suggestion.

Some sentences are repetitive with the abstract in the introduction. Consider rephrasing or presenting the information differently to maintain reader engagement. After explaining the importance of Trichinella spp. detection, move more fluidly into the importance of quality assurance in laboratories, making the transition smoother for readers. Give a brief overview of previous studies or existing systems before diving into specifics. This will help to contextualize the study. Expand on why the magnetic stirrer artificial digestion technique is considered the gold standard method. What makes it superior to other methods? When mentioning the EU data of 117 human cases, provide more context. Over what period were these cases reported? This could provide clarity on the current state of the problem. While mentioning various EU regulations and directives, provide a brief explanation or importance of each. Expand on what is meant by "in other substrates of human interest," this is ambiguous and might confuse readers unfamiliar with the topic.

Response: We followed reviewer’s suggestions.

The methodology section could benefit from improved organization and clarity. Consider breaking down the section into subsections or using bullet points to make it easier for readers to follow the steps of your research. Provide a more detailed explanation of what Satellite Laboratories (SLs) are and the Quality Management System (QMS) based on UNE/EN ISO 17025. This would help readers who may not be familiar with these terms to understand the context better. When discussing the selection criteria for the SLs, explain in more detail how the quantitative and qualitative factors were weighted or evaluated to select the laboratories. This will provide transparency in the selection process. The table describing the activities of each satellite laboratory (Table 1) could be formatted more clearly. Consider using a table format that includes headings for "Satellite Laboratory" and "Activity" to make it easier to read. In the section on "Deviation classification and Coding," provide examples or explanations for each type and subtype of deviation (e.g., Type 1, Subtype (a)). This will help readers understand the significance of each deviation type.

Response: We followed reviewer’s suggestions.

Provide more context and explanation for the statistical analyses performed, such as Canonical Discriminant Analysis (CDA). Explain why CDA was chosen and how it relates to the research objectives.

Response: We followed reviewer’s suggestion.

When discussing multicollinearity, explain the potential impact of multicollinearity on the results and why it is important to address it. Mention any actions taken to mitigate multicollinearity. In the section on canonical coefficients and loading interpretation, provides more information on how these coefficients were interpreted and their relevance to the study. Explain how the results were spatially represented, especially when discussing the creation of a territorial map. Provide details on the methodology used for this representation. Elaborate on the cross-validation procedure and its significance in assessing the accuracy of the discriminant functions. Explain the practical implications of the classification accuracy and Press' Q statistic. Ensure that all relevant references are cited, especially when referring to regulations or standards (e.g., Regulation EU 2015/1375).

Response: We followed reviewer’s suggestion.

Ensure that the presentation of results is clear and well-structured. Use subheadings, bullet points, or tables to organize the information for easier comprehension. Provide a brief explanation of the statistical tests used and their relevance to the analysis. This will help readers understand the significance of the reported values (e.g., VIF, F-statistic). While you've reported numerical results, add some interpretation to help readers understand the practical implications of the findings. What do these results mean in the context of the study's objectives? Consider adding visual aids (e.g., graphs, charts) to illustrate key findings, especially if the data is complex. Visual representation can make the results more accessible.

Response: We followed reviewer’s suggestion.

Start the discussion section by summarizing the key findings from the results section. Provide a concise overview of what the data revealed. Discuss how your findings align or differ from previous research in the same field. Highlight any new insights or contributions your study brings to the existing knowledge. Clearly state the implications of your findings. How do they relate to the broader context of meat inspection, Trichinella spp. control, and quality assurance in satellite laboratories? Acknowledge the limitations of your study. What constraints or challenges did you encounter during the research, and how might they have influenced the results? Offer practical recommendations based on your findings. What steps can be taken to address the issues identified in the satellite laboratories? How can the discrimination tool be improved or utilized effectively? Summarize the main points of the discussion and reiterate the significance of your study's contribution to the field of meat inspection and laboratory quality assurance.

Response: We followed reviewer’s suggestio

Ensure consistent use of terms. For example, decide whether to use "Discriminant Canonical Analysis" or "Canonical Discriminant Analysis." While they aren't presented here, ensure that any figures or tables in the article are well-integrated and referenced in the text. Ensure that all claims or data points have appropriate and recent citations. There are instances where sentences can be shortened or made more concise, which would improve readability.

Response: We followed reviewer’s suggestions

Remember, constructive feedback should be taken in the spirit of improving the quality and readability of the article. These suggestions are to aid in that process.

Response: We thank the reviewer for his/her attention and dedication for his/her evaluation of our work.

Reviewer 2 Report

Comments and Suggestions for Authors

Dear Authors, 

Your manuscript is well written, but I don't understand its purpose. Given that the control of Trichinella infection in animals is very well regulated at the EU level (see COMMISSION IMPLEMENTING REGULATION (EU) 2015/1375 of 10 August 2015, laying down specific rules on official controls for Trichinella in meat), personally, the manuscript, with all the statistical analyses and mathematical interpretations, seems to me just a reinvention of the wheel! Moreover, you can check the International Commission on Trichinellosis recommendations on the ICT website (http://www.trichinellosis.org/Guidelines.html) to see how well this activity is regulated.

Additionally, suppose your final conclusion (The study highlights deficiencies in technique procedures, necessitating measures for result reliability due to facilities' unfamiliarity with extensive QMS documentation) is really true. In that case, those facilities shouldn't be accredited for Trichinella control! As such, your study is importance-less.

Comments on the Quality of English Language

English is fine, maybe small adjustments are needed.

Author Response

Comments and Suggestions for Authors

Dear Authors, 

Your manuscript is well written, but I don't understand its purpose. Given that the control of Trichinella infection in animals is very well regulated at the EU level (see COMMISSION IMPLEMENTING REGULATION (EU) 2015/1375 of 10 August 2015, laying down specific rules on official controls for Trichinella in meat), personally, the manuscript, with all the statistical analyses and mathematical interpretations, seems to me just a reinvention of the wheel! Moreover, you can check the International Commission on Trichinellosis recommendations on the ICT website (http://www.trichinellosis.org/Guidelines.html) to see how well this activity is regulated.

Additionally, suppose your final conclusion (The study highlights deficiencies in technique procedures, necessitating measures for result reliability due to facilities' unfamiliarity with extensive QMS documentation) is really true. In that case, those facilities shouldn't be accredited for Trichinella control! As such, your study is importance-less.

Response: We appreciate reviewer’s feedback and careful consideration of the manuscript. However, even if we appreciate his/her concerns about the manuscript's purpose and relevance, we think there have been misunderstanding issues regarding the meaning of the study and how it was implemented and would like to provide some context to address them.

While it is true that the control of Trichinella infection in animals is indeed well-regulated at the EU level and that there are established regulations and guidelines in place, this research aimed to shed light on the practical implementation and challenges faced by satellite laboratories in adhering to these regulations. The manuscript's focus was to analyze the on-ground execution and quality assurance aspects of Trichinella spp. testing, especially in satellite laboratories that may not have full accreditation. In doing so, we aimed to identify the real-world obstacles that might hinder compliance with these regulations despite their existence. By highlighting these practical challenges and variations in the application of regulations, we sought to underscore the importance of continuous improvement and oversight in the field of meat inspection, even when regulations are in place.

Regarding the recommendation that certain facilities should not be accredited for Trichinella control, the intention was not to dismiss these facilities but rather to underscore the need for targeted improvements and additional support to ensure their full compliance. Sometimes, shortcomings in adherence to regulations can be attributed to a lack of resources, inadequate training, or other operational factors. Identifying these issues provides an opportunity for addressing them and enhancing the overall quality of Trichinella control processes.

In essence, the study aimed to provide insights into the practical challenges faced by satellite laboratories in meeting regulatory standards, which, despite existing regulations and guidelines, can vary in their implementation and effectiveness. The ultimate goal is to improve the quality of Trichinella control procedures, even in well-regulated environments, by addressing real-world obstacles.

This has aready been performed in other high quality research examples published in JCR indexed journals, but has not been done for the area and topic that our study deals with, hence, suggesting the need for its developement, and sort of the necessity leading the materilization of this work.

For instance, these are examples of papers published in 2023 about the relevance of testing the adherence of laboratories to the respective regulation of each particular situation, which as these studies conclude should be encouraged, not discredited.

Aybar Espinoza, M. S., Flink, C., Boisen, N., Scheutz, F., & Käsbohrer, A. (2023). Microbiological sampling and analyses in the food business operators’ HACCP-based self-control programmes. Frontiers in Food Science and Technology3, 1110359.

Miller, W. G., Myers, G., Cobbaert, C. M., Young, I. S., Theodorsson, E., Wielgosz, R. I., ... & Gillery, P. (2023). Overcoming challenges regarding reference materials and regulations that influence global standardization of medical laboratory testing results. Clinical Chemistry and Laboratory Medicine (CCLM), 61(1), 48-54.

Basbug, Gokce, Ayn Cavicchi, and Susan S. Silbey. "Rank Has Its Privileges: Explaining Why Laboratory Safety Is a Persistent Challenge." Journal of Business Ethics 184.3 (2023): 571-587.

Cabral, S. M., Harris, A. D., Cosgrove, S. E., Magder, L. S., Tamma, P. D., & Goodman, K. E. (2023). Adherence to Antimicrobial Prophylaxis Guidelines for Elective Surgeries Across 825 US Hospitals, 2019–2020. Clinical Infectious Diseases, 76(12), 2106-2115.

To this end, we thank you for your time and attention and for considering this manuscript.

Round 2

Reviewer 1 Report

Comments and Suggestions for Authors

include the need for a clearer and more concise introduction in the abstract, addressing awkward phrasing, avoiding repetition, and enhancing the organization of the methodology section.

Comments on the Quality of English Language

Moderate editing of English language required

Reviewer 2 Report

Comments and Suggestions for Authors

Dear Authors,

There are many discussions, interpretations, adjustments, etc., the legislation being very clear: it is or is not accredited! A lot of mathematical analysis, what to do, how to do it, etc., when European legislation is straightforward!...a kind of beating the water in the mill machine!

However, I recommended acceptance of the manuscript.